# Growth, Stratification, and Liberation of Phosphorus-Rich C_2_S in Modified BOF Steel Slag

**DOI:** 10.3390/ma13010203

**Published:** 2020-01-03

**Authors:** Lei Rao, Yuanchi Dong, Mancheng Gui, Yaohui Zhang, Xingmei Shen, Xingrong Wu, Fabin Cao

**Affiliations:** 1School of Metallurgical and Ecological Engineering, University of Science and Technology Beijing, Beijing 100083, Chinadychi@ahut.edu.cn (Y.D.); 2Ma’anshan Iron and Steel Co., Ltd., Ma’anshan 243002, China; guimancheng@126.com (M.G.); zyaohui@sina.com (Y.Z.); 3AnHui Province Key Laboratory of Metallurgy Engineering & Resources Recycling, Anhui University of Technology, Ma’anshan 243002, China

**Keywords:** steel slag, C_2_S, phosphorus, stratification, liberation degree

## Abstract

Basic oxygen furnace (BOF) slag was modified by adding 3.5% SiO_2_ and holding at 1673 K for 0, 5, 40, 90, 240, or 360 min. Kilo-scale modification was also carried out. The growth, stratification, and liberation of P-rich C_2_S in the modified slag were investigated. The optimum holding time was 240 min, and 90% of C_2_S grains were above 30 μm in size. The phosphorus content increased with holding time, and after modification, the phosphorus content in C_2_S was nearly three times higher than that in the original slag (2.23%). Obvious stratification of C_2_S was observed in the kilo-scale modification. Upper C_2_S particles with a relatively larger size of 20–110 μm was independent of RO (FeO-MgO-MnO solid solution) and spinel, which is favorable for liberation. Lower C_2_S was less than 3 μm and was embedded in spinel, which is not conducive to liberation. The content of phosphorus in upper C_2_S (6.60%) was about twice that of the lower (3.80%). After grinding, most of the upper C_2_S existed as free particles and as locked particles in the lower. The liberation degree of C_2_S in the upper increased with grinding time, from 86.02% to 95.92% in the range of 30–300 s, and the optimum grinding time was 180 s. For the lower slag grinding for 300 s, the liberation degree of C_2_S was 40.07%.

## 1. Introduction

Nowadays, the utilization of steel slag can be divided into two aspects. One is internal recycling of the slag in the steel-making process. The other is the external recycling of it as building materials, agricultural fertilizer, functional materials, etc. [1,2,3,4,5,6]. The internal recycling of steel slag is considered to be the best method because calcium, magnesium, manganese, vanadium, iron, rare elements, etc., in the slag can be recovered, and a large portion of slagging agents, such as limestone and fluorite, can be saved [7,8,9]. However, phosphorus, which is a harmful element to steel, greatly limits the internal recycling of steel slag. Therefore, removal of phosphorus from steel slag has attracted the attention of researchers. Ono et al. [10] removed phosphorus from steel slag by floating P-containing dicalcium silicate (C_2_S), owing to its different specific gravity. Kubo et al. [11] adopted magnetic separation to remove phosphorus-enriched phases from multiphase dephosphorization slag. Li et al. [12] investigated the centrifugal enrichment of P-concentrated phases from CaO-SiO_2_-FeO-MgO-P_2_O_5_ melt. Wang et al. [13] studied the enrichment behavior of phosphorus in CaO-SiO_2_-FeO_x_-P_2_O_5_ molten slag, and they pointed out that a higher temperature is favorable for phosphorus enrichment from molten slag to C_2_S particles. Selectively extracting phosphorus from steel slag through a leaching process was carried out by Teratoko et al. [14], Numata et al. [15], and Du et al. [16,17,18,19,20,21].

Using either physical or chemical methods, the distribution of phosphorus in steel slag is important for its separation, not only in relation to separation efficiency but also for the phosphorus grade in P-containing phases. The content of phosphorus in steel slag is relatively low, in the range of 1%–3%, and it is mainly distributed in C_2_S, tricalcium silicate (C_3_S), and, to a lesser extent, in the matrix [8,22,23]. The structure of C_3_S is loose, and after grinding it becomes small particles and adheres to the surface of magnetic particles, which hinders separation. In our previous studies, it was found that almost all of the phosphorus was enriched in C_2_S by adding 3.5 wt% SiO_2_ to modify steel slag, and no C_3_S was formed [23]. Moreover, the crystal size of P-rich C_2_S grew faster at 1673 K [24], and after magnetic separation, phosphorus in the magnetic fraction mainly came from unliberated C_2_S [25]. It is, thus, clear that the degree of liberation of P-rich C_2_S is the key factor affecting the efficiency of separation. However, few studies have focused on this factor.

In this work, basic oxygen furnace slag (BOF slag) was modified by adding 3.5 wt% SiO_2_ and holding at 1673 K for different times, and kilo-scale modification was also carried out. Growth and stratification of P-rich C_2_S in the modified slag was analyzed. After grinding for different times, the respective liberation degrees of C_2_S in the upper and lower modified slags were investigated.

## 2. Experimental

### 2.1. Modification of Original BOF Slag

The original BOF slag was crushed and ground to pass a 200 mesh, and the chemical composition of the slag is listed in Table 1. The slag powder (11 g), mixed with 3.5 wt% SiO_2_, was added into a small MgO crucible and heated at 1793 K for 10 min to melt the mixture. Then, the molten mixture was cooled to 1673 K at the cooling rate of 1 K/min. After holding for 0, 5, 40, 90, 240, or 360 min at 1673 K, the mixture was quickly quenched with water, and the modified slag was obtained.

### 2.2. Kilo-Scale Modification of the BOF Slag

To investigate the stratification of P-rich C_2_S in the modified slag, kilo-scale modification of the slag was carried out. The slag powder (900 g), mixed with 3.5 wt% SiO_2_, was added into a larger MgO crucible (100 mm diameter × 110 mm height) and heated at 1793 K for 10 min. Then, the molten mixture was cooled to 1673 K at the cooling rate of 1 K/min. After holding for 240 min at 1673 K, the mixture was quickly quenched with water. Finally, the quenched, modified slag was cut vertically along the longitudinal direction, and the upper and lower slag samples were taken for the next process.

### 2.3. Liberation of P-Rich C_2_S in the Upper and Lower Modified Slags

The obtained upper and lower slags were ground for 30, 180, or 300 s. Then, the above slag powders were respectively mixed with phenolic resin powder and hot pressed. After polishing and cleaning, the samples were taken for SEM (scanning electron microscope) analysis.

### 2.4. Characterization

The crystal structure of the slags was characterized by a Bruker AXS-D8 Advance X-ray powder diffraction (XRD) under Cu Kα radiation (λ = 0.15405 nm) operated at 40 kV and 40 mA. Micromorphology of the slags was observed by a Philips XL-30 scanning electron microscope and energy dispersive spectrometer (SEM–EDS). The grain size of C_2_S was calculated quantitatively by Photoshop software (Adobe Photoshop CS4, Adobe Systems Inc., San Jose, CA, USA) and NIH image J software (NIH Image public domain software, US National Institute of Health, Bethesda, MD, USA). Particle size distribution of the ground slags was carried out using a RISE-2006 laser particle size analyzer.

## 3. Results and Discussion

### 3.1. Crystal Growth of P-Rich C_2_S in the Modified Slag

Figure 1 shows the SEM images of the modified slag with different holding times at 1673 K, and the EDS results are listed in Table 2. From the figure, gray C_2_S, black RO (FeO-MgO-MnO solid solution), and light gray matrix phases were observed, and the EDS analysis indicated that no phosphorus was found in RO and matrix phases, which means all phosphorus was enriched in C_2_S. As can be seen in Figure 1a,b, many small C_2_S grains can be observed. After holding at 1673 K for 40 min, the small grains gradually disappeared, as shown in Figure 1c. Meanwhile, the growth of the remaining C_2_S grains was obvious, and the number of C_2_S grains per unit area decreased significantly. With the increase in holding time, it was found that adjacent C_2_S grains were connected with each other and eventually formed larger grains, shown in Figure 1d–f. Figure 2 shows a frequency histogram of the grain size distribution for C_2_S with different holding times at 1673 K. Generally, a grain size above 30 μm was favorable for the liberation of C_2_S, while below 30 μm it was difficult to liberate. At the beginning (5 min), almost all C_2_S grains were below 30 μm. After 40 min, half of the grains were larger than 30 μm. The grains above 30 μm reached 90% when the holding time reached 240 min. At 360 min, the situation was similar to that of 240 min, except that 60–110 μm grains were more numerous than those at 240 min. Considering the energy consumption, 240 min was the optimum holding time.

Note that the ratios of Mg to Fe in the RO phase with different holding times were all greater than 6 (Mg/Fe > 6), which is obviously high. According to Engstrom et al. [26], MgO precipitates earlier in the slag during cooling due to the higher melting point, and FeO and MnO have the same chance to dissolve into MgO, but FeO precipitates earlier than MnO. Thus, it can be concluded that MnO did not precipitate before quenching in this work. Table 3 shows the content of phosphorus in C_2_S with different holding times at 1673 K. It can be seen that, considering reasonable error, the phosphorus content increased with holding time, and it was nearly three times higher than that in the original slag (2.23%).

### 3.2. Stratification of P-Rich C_2_S in the Modified Slag

Due to lower specific gravity, C_2_S would float through the molten slag and lead to stratification [10]. In Section 3.1, no obvious stratification was found, owing to the wall effect. To investigate the stratification of P-rich C_2_S, kilo-scale modification of the slag was carried out at 1673 K for 240 min. Figure 3 shows the XRD patterns of the upper and lower modified slags on a kilo-scale. It can be seen from the figure that C_2_S, RO, and the spinel were the main phases in the upper slag. Besides these, a few weak calcium ferrite (C_2_F) peaks can be observed. According to thermodynamic prediction calculations correlated with experimental results by Gautier et al. [27], C_2_F crystallizes at 1333 K in BOF slag. Thus, due to the high quenching temperature (1673 K) in this work, the thermodynamic conditions for crystallization were not sufficient for C_2_F. For the lower slag, C_2_S and the spinel were the main phases. Note that the intensity of C_2_S in the upper was higher than that in the lower, and the relative intensity of C_2_S and spinel in the upper was obviously higher than that of the lower, which indicates a preferential growth of C_2_S over other phases in the upper.

Figure 4 shows the SEM images of the upper and lower modified slags on the kilo-scale, and the EDS results are listed in Table 4. For the upper, C_2_S was spherical with a grain size in the range of 20–110 μm, and several holes can be observed (Figure 4a). Due to volume expansion caused by crystal transformation of C_2_S during cooling, microcracks occurred on C_2_S, which can be seen clearly in Figure 4b. The RO phase was about 20–40 μm, and spinel was dispersed in the matrix, forming a strip shape with a smaller size. These phases in the upper slag were independent of each other, and the interface between each phase was clear, which was favorable for the liberation of C_2_S. For the lower slag, the micromorphology of the slag was obviously different from that of the upper. C_2_S particles in the lower slag were much smaller (less than 3 μm) than those in the upper slag, and these small grains were arranged in a flower pattern, as shown in Figure 4c. Note that C_2_S was embedded in spinel, which can be seen clearly in Figure 4d, and it was not conducive to the liberation of C_2_S.

From Table 4, it can be seen that 9.81 at% Fe dissolved in C_2_S for the lower slag. In the early holding stage, C_2_S precipitated first and grew quickly. At this stage, C_2_S mainly grew by itself, controlled by mass transfer, and the viscosity of the slag was relatively lower, which was conducive to the diffusion of ions and the floating of C_2_S. C_2_S floated above the molten slag, and this led to the decrease of Ca and Si in the lower part of molten slag. Thus, in the middle holding stage, C_2_S precipitated with more Fe, Mg, etc., in the lower slag and grew slowly, owing to the reduction of Ca and Si. This is consistent with the XRD results in Figure 3. In the late holding stage, the growth of C_2_S in the upper was mainly due to the merging of grains, according to Figure 1d,e. Due to dissolving of Fe, Mg, etc., the specific gravity of C_2_S in the lower was relatively higher, which prevented floating, and they were mostly embedded in spinel. Moreover, the increased viscosity of the molten slag was another reason for preventing floating. Table 5 shows the content of phosphorus in the upper and lower C_2_S, which was calculated from the atomic ratios in Table 4. It can be seen that the content of phosphorus in upper C_2_S (6.60%) was about two times higher than that in lower C_2_S (3.80%).

### 3.3. Liberation Degree of P-Rich C_2_S in the Modified Slag

The above studies proved that the micromorphology of the upper modified slag was conducive to the liberation of C_2_S, while that of the lower slag was not. Then, the upper and lower slags were ground. The particle size distribution with different grinding times is shown in Figure 5. From Figure 5a, it can be seen that the average particle size of the upper slag was 10.92 μm after grinding for 30 s and 8.12 μm after 180 s. When the grinding time reached 300 s, the average particle size was 7.84 μm, and the size distribution curve almost coincided with that of 180 s, which indicates that prolonging the grinding time after 180 s had no obvious effect on the particle size. The particle size distribution of the lower slag after grinding for 300 s is shown in Figure 5b. As can be seen in the figure, the average particle size was 8.22 μm, which was larger than that of the upper slag. This indicates that the lower slag was more difficult to grind than the upper, which may be due to more Fe-bearing phases.

Figure 6. SEM images of the upper modified slag after different grinding times. It can be seen from Figure 6 that C_2_S existed both as free particles and locked particles, most of which existed in the form of free particles. According to the classification of locked particles, the C_2_S locked particles in the upper slag belonged to the adjacent type. Here, the spinel and the matrix are collectively referred to as the M phase, and for the RO phase, they locked with the M phase. SEM images of the lower slag after grinding for 300 s are shown in Figure 7. It can be seen that most of C_2_S existed in the form of locked particles with the M phase, and they are mostly of the encapsulated type.

After respectively grinding the upper and lower slags, the liberation degree of C_2_S was determined by SEM. The liberation degree of a mineral (*L*) is defined by Equation (1):
(1)$$L=\frac{q_{f}}{q_{f}+q_{l}}\times 100\%$$
in which *q_f_* is content of free particles, and *q_l_* is content of locked particles. In this work, the particle counting method was used to calculate the liberation degree of C_2_S, and the calculation Equation is as follows:
(2)$$L_{C_{2}S}=\frac{N_{f\left(C_{2}S\right)}}{N_{f\left(C_{2}S\right)}+\frac{3}{4}N_{l\left(3/4\right)}+\frac{1}{2}N_{l\left(1/2\right)}+\frac{1}{4}N_{l\left(1/4\right)}+\frac{1}{8}N_{l\left(1/8\right)}+\frac{1}{16}N_{l\left(1/16\right)}}$$
in which *N_f_* is the number of C_2_S free particles, and *N_l_* is the number of C_2_S locked particles. The liberation degree of C_2_S in the upper and lower modified slags as calculated by Equation (2) is listed in Table 6. As can be seen in the table, the liberation degree of C_2_S in the upper slag increased with grinding time of the modified slag, from 86.02% to 95.92% in the range of 30–300 s. For the lower slag ground for 300 s, the liberation degree of C_2_S was only 40.07%. Therefore, considering energy consumption, 180 s was the optimum grinding time for liberation of C_2_S from the upper slag.

## 4. Conclusions

Growth, stratification, and liberation of P-rich C_2_S in the modified BOF slag were investigated, and the conclusions are as follows:

(1) The optimum holding time at 1673 K to modify the slag was 240 min, after which 90% of C_2_S grains were above 30 μm. The phosphorus content increased with holding time, and after modification, the phosphorus content in C_2_S was nearly three times higher than that in the original slag (2.23%).

(2) Obvious stratification of C_2_S was observed in the kilo-scale modification. Upper C_2_S with a relatively larger size of 20–110 μm was independent of RO and spinel, which is favorable for liberation. The lower C_2_S was less than 3 μm and was embedded in spinel, which is not conducive to liberation. The phosphorus content in upper C_2_S (6.60%) was about two times higher than that in the lower (3.80%).

(3) After grinding, most of the C_2_S existed as free particles in the upper and as locked particles in the lower. The liberation degree of C_2_S in the upper increased with grinding time, from 86.02% to 95.92% in the range of 30–300 s, and the optimum grinding time was 180 s. For the lower slag ground for 300 s, the liberation degree of C_2_S was 40.07%.

## Figures and Tables

**Figure 1 materials-13-00203-f001:**
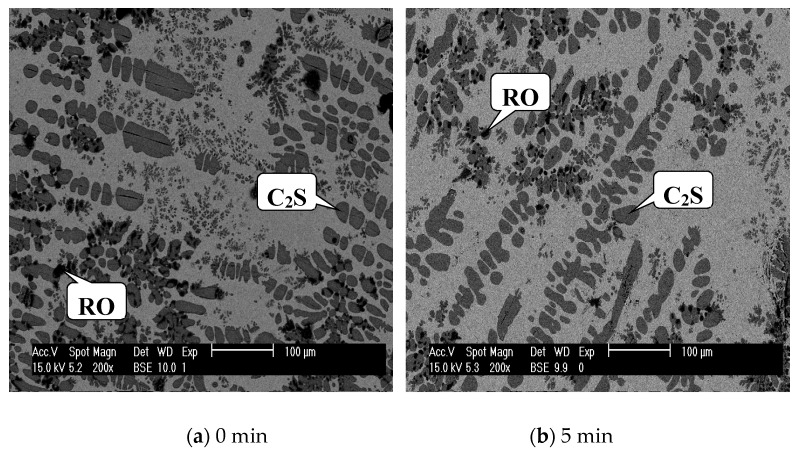
SEM images of the modified slag with different holding times at 1673 K.

**Figure 2 materials-13-00203-f002:**
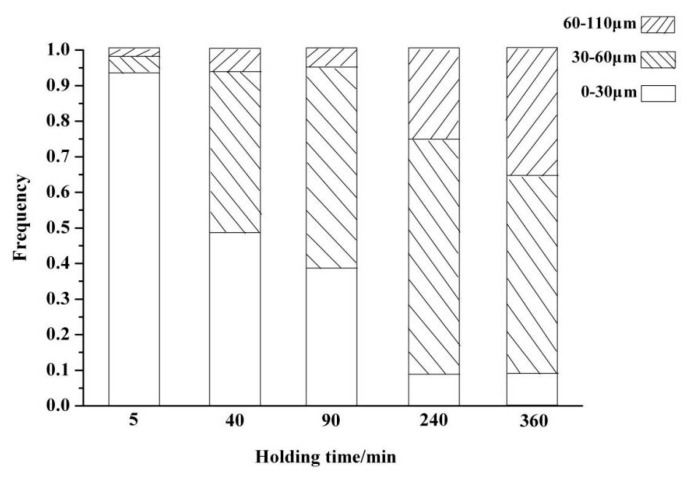
Frequency histogram of the grain size distribution for C_2_S with different holding times at 1673 K.

**Figure 3 materials-13-00203-f003:**
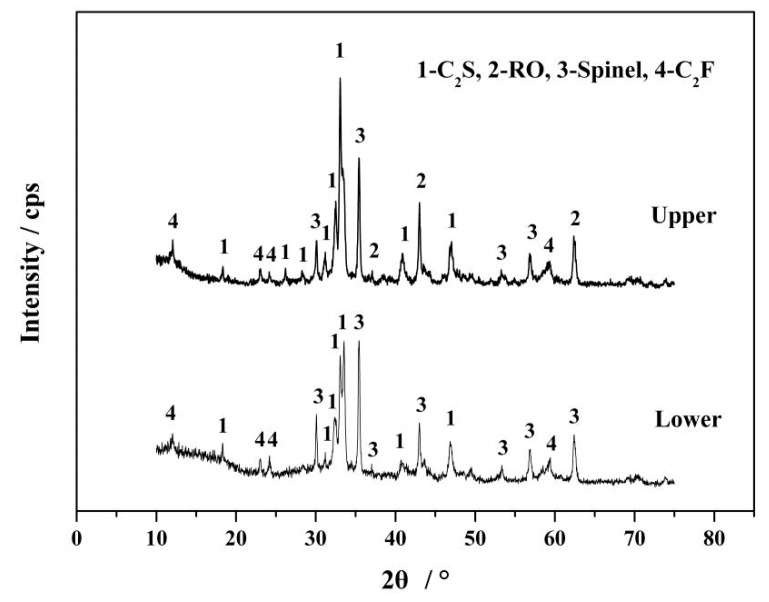
XRD patterns of the upper and lower modified slags held at 1673 K for 240 min on a kilo-scale.

**Figure 4 materials-13-00203-f004:**
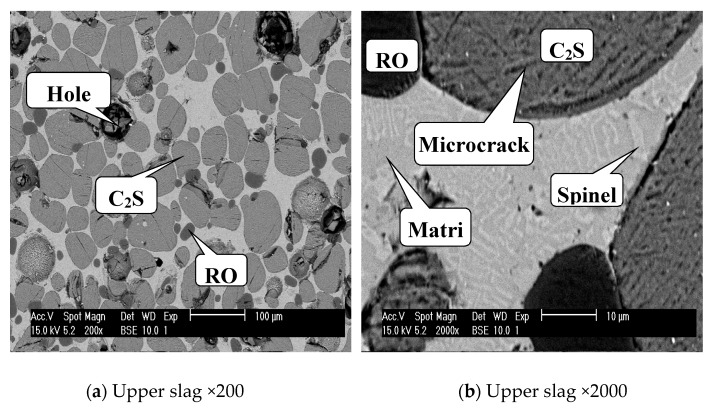
SEM images of the upper and lower modified slags held at 1673 K for 240 min after kilo-scale processing.

**Figure 5 materials-13-00203-f005:**
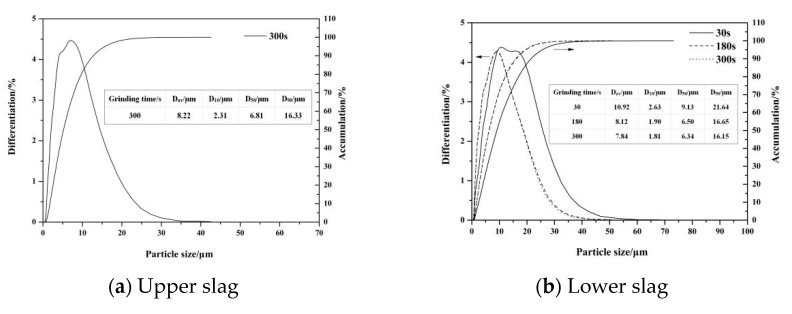
Particle size distribution of the upper and lower modified slags with different grinding times.

**Figure 6 materials-13-00203-f006:**
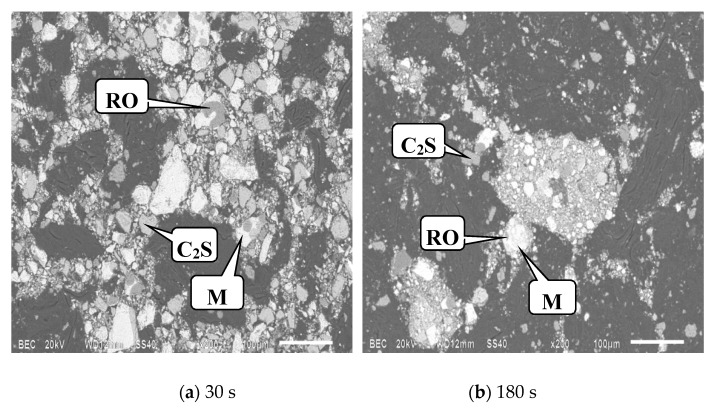
SEM images of the upper modified slag after different grinding times.

**Figure 7 materials-13-00203-f007:**
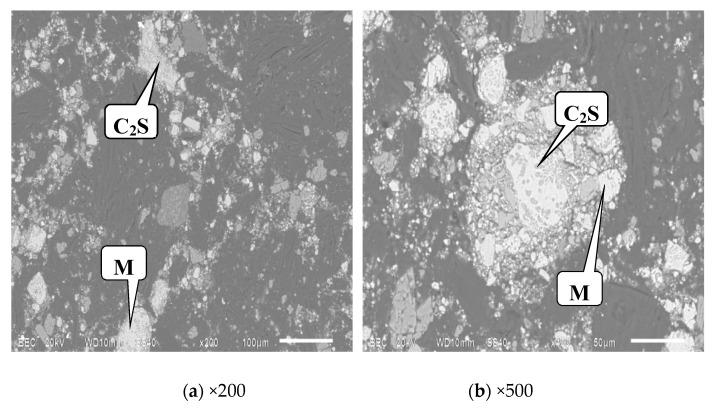
SEM images of the lower modified slag after grinding for 300 s.

**Table 1 materials-13-00203-t001:** Chemical composition of the original BOF slag.

Sample	CaO	SiO_2_	P_2_O_5_	MgO	MnO	Al_2_O_3_	TiO_2_	FeO	T.Fe
Original slag	43.42	11.06	2.23	8.82	1.65	1.58	0.59	15.27	21.98

**Table 2 materials-13-00203-t002:** EDS analysis of the modified slag with different holding times at 1673 K (at%).

Holding Time/min	Phase	Ca	Si	Fe	Mg	Al	Mn	P	O
0	C_2_S	31.70	16.99	-	1.09	-	-	2.73	47.49
	RO	-	-	7.11	58.76	-	-	-	34.13
	Matrix	16.70	7.56	24.08	8.10	2.17	2.29	-	38.19
5	C_2_S	39.86	17.08	-	-	-	-	2.78	40.29
	RO	-	-	8.78	56.91	-	-	-	34.31
	Matrix	22.68	4.41	26.55	3.88	2.24	2.03	-	38.20
40	C_2_S	38.68	16.01		1.03	-	-	2.94	41.33
	RO	-	-	8.15	56.60	-	-	-	35.24
	Matrix	22.88	4.98	26.6	4.17	2.29	2.18	-	36.90
90	C_2_S	39.76	17.01	-	1.02	-	-	3.17	39.04
	RO	-	-	7.80	57.13	-	-	-	35.08
	Matrix	24.25	5.06	26.39	4.28	2.16	1.92	-	35.95
240	C_2_S	38.98	17.21		1.32	-	-	3.23	39.27
	RO	-	-	6.26	58.53	-	-	-	35.21
	Matrix	22.67	5.07	26.47	4.26	2.25	1.94	-	37.32
360	C_2_S	39.88	16.73	-	1.02	-	-	3.52	38.85
	RO	-	-	9.06	57.85	-	-	-	33.09
	Matrix	23.70	4.98	29.50	4.27	2.09	2.85	-	32.60

**Table 3 materials-13-00203-t003:** Content of phosphorus in C_2_S with different holding times at 1673 K.

Holding Time/min	0	5	40	90	240	360	Original Slag
P_2_O_5_ (C_2_S)/wt%	5.69	5.71	6.18	6.41	6.56	7.08	2.23

**Table 4 materials-13-00203-t004:** EDS analysis of the upper and lower modified slags held at 1673 K for 240 min processed at the kilo-scale (at%).

Slag	Phase	Ca	Si	Fe	Mg	Al	Mn	P	O
Upper	C_2_S	36.59	17.10	-	0.86	-	-	3.08	42.37
	RO	-	-	7.97	57.23	-	-	-	34.80
	Spinel	5.39	1.12	37.81	12.18	3.99	2.78	-	36.73
	Matrix	28.27	4.71	26.03	0.90	3.20	1.29	-	35.62
Lower	C_2_S	25.30	12.10	9.81	3.73	1.35	-	1.48	46.24
	Spinel	6.44	2.87	35.41	12.25	1.81	3.48	-	37.74
	Matrix	28.86	4.56	23.86	1.07	1.88	1.93	-	36.81

**Table 5 materials-13-00203-t005:** Content of phosphorus in the upper and lower C_2_S held at 1673 K for 240 min processed at the kilo-scale.

Sample	Upper	Lower	Original Slag
P_2_O_5_ (C_2_S)/wt%	6.60	3.80	2.23

**Table 6 materials-13-00203-t006:** Liberation degree of C_2_S in the upper and lower modified slags.

Modified Slag		Number of C_2_S Free Particles	Number of C_2_S Locked Particles						Number of Other Particles	Total Number of Observed Particles	Liberation Degree of C_2_S/%
			3/4	1/2	1/4	1/8	1/16	<1/16			
Upper	30 s	150	23	8	8	7	4	0	143	343	86.02
	180 s	103	4	1	6	1	0	0	100	212	95.12
	300 s	97	0	5	4	4	2	0	136	248	95.92
Lower	300 s	40	38	28	30	35	27	0	38	251	40.07

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
