# Peer review of "Growth, Stratification, and Liberation of Phosphorus-Rich C2S in Modified BOF Steel Slag"

_materials, 2020, doi:10.3390/ma13010203_

Round 1

Reviewer 1 Report

Present manuscript describes the modification of BOF steel slag by SiO2 and growth phosphorous-rich Ca2SiO4 (C2S). This article is of medium interest. The extensive editing of English editing and style is required. This must be attended to.   

However, there are some concerns that need to be addressed:

All abbreviation used in text should be explained at the first appearance g. Basic oxygen furnace slag (BOF slag), di-calcium silicate Ca2SiO4 (C2S) and tricalcium silicate Ca3SiO5 (C3S) A thorough revision of language and style, preferably by a native speaker or language service, is required. I have noted many deficiencies in language and grammar, for example: 1 line 30-33 – “The internal recycling of steel slag is considered to be the best way [7-9], because calcium, magnesium, manganese, vanadium, iron, rare element, etc., in the slag can be recovered, and a large amount of slagging agents, such as limestone and fluorite, can be saved.” 1 line 33 – “However, phosphorous, which is a harmful element for steel…”. 2 line 52-53 – “However, few studies have focused on it.” – here also statement without proper citation 5 line 130-131 “According to thermodynamic prediction calculations correlated with experimental results by…” 10 line 226-227 “After grinding, most of C2S exist as free particles in the upper, while as locked particles in the 226 lower.” The XRD pattern of “Upper” and “Lower” modified slags is almost identical, the small difference could be explained by presence of admixture. In according to Materials policy, the input of which authors should be specified. The fonts in the Figure 5 seems to be deformed – please correct it.

Author Response

The reviewer provided valuable comments on our manuscript, and we would like to express our appreciation for his or her advice. Our responses to the questions and comments are provided below. In addition, all of the revised parts in the manuscript have been highlighted in red font.

Reviewer #1

(1) All abbreviation used in text should be explained at the first appearance.

Answers: Thank you for your suggestion. All abbreviations have been explained at the first appearance: dicalcium silicate-C2S, basic oxygen furnace slag-BOF, tricalcium silicate-C3S, FeO-MgO-MnO solid solution-RO, calcium ferrite-C2F.

(2) A thorough revision of language and style, preferably by a native speaker or language service, is required. And here also statement without proper citation.

Answers: The language of this manuscript has been revised by MDPI English Editing Service. And line 30-33 statement citation has been revised.

(3) The XRD pattern of “Upper” and “Lower” modified slags is almost identical, the small difference could be explained by presence of admixture.

Answers: As the reviewer said, the XRD pattern of the upper and lower slags is almost identical. The analysis for main phases was combined with SEM-EDS results (Figure 4 and Table 4). C2S, RO, and the spinel were the main phases in the upper slag, and C2S and the spinel were the main phases in the lower slag. Besides, the intensity of C2S, and the relative intensity of C2S and spinel in the upper were higher than those in the lower, which indicates a preferential growth of C2S over other phases in the upper.

(4) The fonts in the Figure 5 seems to be deformed – please correct it. 

Answers: Figure 5 has been revised.

Reviewer 2 Report

this is a very interesting paper providing significant results on the Growth, stratification and liberation of phosphorus3 rich C2S in modified BOF steel slag. It is of fundamental interest, and should hence be published.

HOwever, I would suggest author to improve abstract of this paper to make to highlight the importance of this work and conclusion. 

Author Response

The reviewer provided valuable comments on our manuscript, and we would like to express our appreciation for his or her advice. Our responses to the questions and comments are provided below. In addition, all of the revised parts in the manuscript have been highlighted in red font.

Reviewer #2

(1) I would suggest author to improve abstract of this paper to make to highlight the importance of this work and conclusion. 

Answers: Thank you for your suggestion. The abstract has been revised.